# Effect of Porosity on Functional Properties of Lead-Free Piezoelectric BaZr_0.15_Ti_0.85_O_3_ Porous Ceramics

**DOI:** 10.3390/ma13153324

**Published:** 2020-07-26

**Authors:** Lavinia Curecheriu, Vlad Alexandru Lukacs, Leontin Padurariu, George Stoian, Cristina Elena Ciomaga

**Affiliations:** 1Dielectrics, Ferroelectrics & Multiferroics Group, Faculty of Physics, Alexandru Ioan Cuza University, 11 Carol I Blvd., 700506 Iasi, Romania; vlad.lukacs@uaic.ro (V.A.L.); leontin.padurariu@uaic.ro (L.P.); 2National Institute of Research and Development for Technical Physics, 700050 Iasi, Romania; gstoian@phys-iasi.ro; 3Research Department, Faculty of Physics, Alexandru Ioan Cuza University, 11 Carol I Blvd., 700506 Iasi, Romania; cristina.ciomaga@uaic.ro

**Keywords:** porous ceramics, dielectric properties, DC tunability, piezoelectric coefficient

## Abstract

The present paper reports the dependence of dielectric, ferroelectric and piezoelectric properties on the porosity level in BaZr_0.15_Ti_0.85_O_3_ ceramics with porosity from 5% to 21%. Microporosity with 0–3 connectivity has been produced using PMMA microspheres as a sacrificial template. The functional properties (dielectric, ferroelectric and piezoelectric effect) are mostly affected by the “dilution effect”: permittivity decreases by 40% when porosity increases by 21%, and P_max_ decreases from 13 to 5 µC/cm^2^ while the P_rem_ is in the range of (2–8) µC/cm^2^. However, the reduction of the zero-field permittivity and hysteretic behaviour of ε(E) while the tunability level is still high makes from porous ceramics interesting materials for tunability application.

## 1. Introduction

Ferroelectric materials represent a significant class of electroactive dielectrics that display high polarisation and permittivity, together with large electromechanical coupling. Their functional properties, expressed through pyroelectricity, piezoelectricity, electro-optic behaviour and tunability, recommend them for applications in sensors, transducers and actuators, energy harvesters and Wi-Fi communications [1,2]. All electroceramics exhibit small amounts of porosity that is usually regarded as a defect related to imperfections in the material during processing. On the other hand, in the last decade, some authors deliberately introduced porosity in order to create composite materials with tailored properties for specific applications [2,3,4]. In ferroelectric and piezoelectric materials, porosity has been taken into consideration at the micro- and nanoscale [5,6] on bulk materials [7,8,9], textured materials [10], aerogels [11] and thin films [12]. One reason for the intentional introduction of porosity was to reduce the acoustic impedance of piezoelectric materials. This reduction improves the match between the piezoelectric material and the fluid environment used in the case of applications like medical ultrasounding, non-destructive fissure testing of mechanical parts and SONAR. Another reason for intentional induced porosity was the reduction of permittivity in ferroelectric materials. Small permittivity is a requirement in piezoelectric, pyroelectric and energy harvesting applications since figures of merit are inversely proportional to the materials permittivity for a given constant value of stress [13,14]. Furthermore, one must take into consideration the influence of porosity on the fatigue behaviour during multiple polarisation reversals in the case of ferroelectric memories and high-strain actuator applications [15]. In order to explain the physical properties of such porous materials, several models have been developed in the last several years. Generally, these models assume that porous ceramics are fully poled in a particular direction. This assumption is correct only in the case of dense bulk materials. In the case of porous materials, where the microstructure reveals pores with significantly lower permittivity than the ferroelectric bulk in which they are embedded, the applied electric field concentrates in these low permittivity pore regions. This concentration leads to an inhomogeneous distribution of the electric field throughout the structure [9,16]. This inhomogeneous field distribution has generated interest in the research community towards using these porous ferroelectric ceramics in applications requiring a high degree of tunability through the exploitation of nonlinear, electric field-dependent dielectric properties [9,17].

In the last several years, the majority of publications associated with the porosity influence on dielectric, piezoelectric and pyroelectric properties refer to lead zirconate titanate-based ceramics [6,7,8,9,10,11,12,13,14,15]. Only a few papers have reported the effect of porosity on the overall polarization–electric field (P–E) behaviour and DC tunability on lead-free porous ceramics [18,19,20]. This topic is of great interest, since BaTiO_3_-based ceramics will replace the lead-based materials in microelectronic applications in a few years. In this context, Zr-doped BaTiO_3_ (BaZr_y_Ti_1-y_O_3_-BZT) received special attention due to its potential applications in tuneable ceramic capacitors and microwave devices applications [21]. Depending on the Zr^4+^ ion concentration in BaTiO_3_ perovskite materials, a shift from classical ferroelectric to a relaxor ferroelectric takes place. BZT performs like a typical ferroelectric material for Zr^4+^ additions lower than *y* = 0.08. When increasing the Zr^4+^ addition (0.10 < *y* ≤ 0.15), BZT ceramics show a diffuse phase transition with a mixture of structural phases (rhombohedral, tetragonal and cubic) at room temperature. For Zr^4+^ concentrations larger than *y* > 0.20, the Curie temperature *(T_C_*) decreases below room temperature, accompanied by an increase of diffuseness of phase transition, thus indicating a mixed ferroelectric–relaxor behaviour [10]. For *y* > 0.25, a ferroelectric–relaxor crossover in BZT takes place [11,12]. One of the most interesting compositions of Zr-doped BaTiO_3_ is *y* = 0.15. For this composition, a mixture of structural phases with a diffuse phase transition has been reported [22,23] and the electrical properties have been extensively investigated. However, the effect of porosity on dielectric, ferroelectric and piezoelectric properties of this composition has not been systematically investigated.

In the present paper, PMMA microspheres were employed as sacrificial templates for inducing 0–3 type porosity in Zr-doped BaTiO_3_ with *y* = 0.15. The dielectric, ferroelectric, piezoelectric and tunability properties are investigated and discussed in relation to microstructure particularities.

## 2. Materials and Methods

### 2.1. Sample Processing and Preparation

Porous Ba(Zr,Ti)O_3_ (BZT) ceramics with *y* = 0.15 composition (BaZr_0.15_Ti_0.85_O_3_) have been prepared by adding 0, 5, 10, 15, 20, 30, 35 and 40 vol. % poly(methyl methacrylate) (PMMA) as a pore-forming agent. Firstly, BZT nanopowders were prepared by solid-state reaction, starting from BaCO_3_ (Solvay, 99.9% purity), TiO_2_ (Toho, 99.9% purity) and ZrO_2_ (99.9%). The raw materials were weighted and mixed with distilled water for 24 h according to the chemical formula. Subsequently with freeze-drying, the powders were calcined at 1100 °C for 4 h in order to promote the solid-state reaction. After calcination, the powders were sieved and manually re-milled. In order to prepare porous ceramics, the obtained BZT nanopowders, with diameters around 300 nm, were mixed in desired proportions (0–40 vol%) with PMMA microspheres, having diameters around 10 µm (Figure 1a) and then milled in acetone for 20 min in order to promote homogeneous mixing. The mixed powders were isostatically pressed at 1500 bar in disks with 10 mm in diameter. The thermal treatment was established based on the thermogravimetric analysis of PMMA reported by Zhang et al. [24]. Therefore, the initial heating rate up to 240 °C was chosen at 2 °C/min, followed by 1 °C/min up to 420 °C and 2 °C/min up to 850 °C, in order to guarantee the complete burnout of PMMA. The green samples were sintered at 1500 °C for 4 h. The density of the sintered ceramics was measured using the Archimedes method and the variation of porosity with the content of the pore-forming agent is shown in Figure 1b. The porosity increases with an increasing content of PMMA at a given sintering temperature. When the content of PMMA increased from 0% to 40%, the porosity increased from 1% to 21%, thus indicating that porosity can be controlled by content of PMMA. These results are in good agreement with those reported for other BaTiO_3_-based ceramics with PMMA addition [24].

### 2.2. Experimental Details 

The phase composition of the sintered ceramics was determined using X-ray diffraction (XRD) with CuKα radiation (Panalytical CubiX), with a scan step of 0.02°, counting time of 7 s/step and 2θ ranging between 20° and 80°. The microstructure of the porous ceramics was observed through high-resolution scanning electronic microscopy with a Carl Zeiss System NEON40EsB (MicroImaging GmbH, Jena, Germany) and the sample density was estimated by using Archimedes’ method. For electric measurements, Ag electrodes were deposited on the plane-parallel polished surfaces of the ceramics, followed by annealing in open air atmosphere at 200 °C for 2 h. The low-field dielectric measurements were carried out at room temperature using Solartron 1260 (Solartron Analytical, Hampshire, UK) for frequencies ranging from 1 Hz to 1 MHz and at temperatures between 20–120 °C using LCR bridge Hameg HM8118 (Rohde & Schwarz GmbH & Co. KG, Munich, Germany). High-field measurements were performed at room temperature on ceramic disks immersed in silicon oil bath. The *P*(*E*) loops were measured using Radiant Precision Multiferroic II Ferroelectric Test System (Radiant Technologies, INC., Albuquerque, New Mexico, USA) on unpoled materials with a frequency of 1 Hz and double bipolar input as the electric signal. The DC tunability was measured at an oscillator frequency of 18 kHz and 1 V amplitude using a function generator coupled with a TREK 30/20A-H-CE amplifier (TREK, New York, USA) [25]. Piezoelectric coefficient *d*_33_ values were measured by a quasi-static *d*_33_ meter (PiezoMeter 320) (Piezotest Pte. Ltd. Singapore) on the poled ceramics at room temperature at 10 kV/cm for 10 min. The piezoelectric measurements were performed after 24 h.

## 3. Results and Discussion

### 3.1. Phase and Microstructural Characterisation

X-ray diffractograms of porous ceramics, shown in Figure 2, reveal the characteristic single-phase perovskite polycrystalline structure. The absence of any secondary phases evidences that the homovalent substitution of Ti^4+^ by Zr^4+^ has resulted in a homogenous solid solution. The crystallite size of the samples corresponding to the highest intensity (110) was calculated and it was found that it increases from 30.45 nm to 49.26 nm, with increasing porosity. For all the ceramic samples, the (200) peak is not split, but asymmetric and broadened. This suggests the coexistence of structural phases and makes it difficult to structurally solve this compound [26]. Furthermore, the intensity of the (200) peak varies in intensity, and these variations may be due to an increase of internal stress in BZT particles, resulting in an elastic compressive volume strain [27].

The SEM micrographs (Figure 3), performed on fresh fractured ceramics, show the role of progressive addition of PMMA on the ceramic microstructures. The ceramics with low porosity (e.g., 5%—Figure 3b) contain two types of pores: fine spherical pores, with diameters of ~1 µm, originated from the densification of starting BZT nanopowders, together with large spherical pores with average size of about 15 µm, induced after burning the PMMA templates. When increasing the amount of PMMA, the small pores are progressively removed; the dense regions increase in size and the large pores become irregular in size and shape, being percolated in various regions (Figure 3e). In the dense regions, the grain size is large (around hundreds of micrometres for the 1% and 5% porosity) and strongly decreases with the increasing addition of PMMA.

### 3.2. Low-Field Dielectric Properties

Room temperature permittivity and dielectric losses are shown in Figure 4a,b. They indicate a porosity dependent gradual decrease of permittivity, regarded as a *sum property* (or dilution of the ferroelectric phase by the presence of pores), from 2000 (1% porosity) to about 1250 (21% porosity) and good dielectric character (tan*δ* below 8%) for all the porous ceramics, in the 1–10^6^ Hz frequency range. All the ceramics show a small dispersion in frequency (~15% difference between permittivity at 1 Hz and permittivity at 1 MHz) except for ceramic with 8% porosity. In this case, the difference between permittivity at 1 Hz and values of permittivity at 1 MHz is ~25%. Furthermore, this ceramic presents the largest dielectric loss. This behaviour may be assigned to Maxwell–Wagner relaxation due to charge inhomogeneities present inside the ceramic volume (air pore–ceramic interfaces, etc.). Similar behaviour was observed in other porous ceramics and was also assigned to interfacial polarisation [9,20,28]. All the samples show smaller losses for high frequencies (>1 kHz). These losses increase at low frequencies (<100 Hz) due to interfacial polarisation. All the ceramics exhibit a maximum in dielectric loss at ~1 kHz that is independent on porosity, with a relaxation time *τ*~2.2 × 10^−4^ s.

The temperature dependence of permittivity for a fixed frequency of 10 kHz (Figure 5a) shows a maximum corresponding to the ferroelectric–paraelectric phase transition, with a Curie temperature between 60–62.5 °C, irrespective of porosity. The porosity level does not alter the temperature corresponding to the permittivity maximum. However, in the case of the 5% porosity ceramic sample, the maximum values of permittivity at the transition point are higher than those of the 1% porosity ceramic sample. The dielectric losses remain below 7% in the investigated temperature range with a maximum at the transition point. The sample with 12% porosity displays the largest losses in the paraelectric phase, while the densest ceramic (1% porosity) exhibits the smallest losses in all investigated temperature ranges.

In the paraelectric state, the dielectric permittivity of a ferroelectric material follows the Curie–Weiss law:(1)$$\frac{1}{\varepsilon }=\frac{T-T_{0}}{C}\;,\;\left(T>T_{m}\right)$$
where *T*_0_ is the Curie–Weiss temperature and *C* is the Curie constant, and both are frequency dependent. *T_m_* is the temperature corresponding to the maximum value of permittivity. In Figure 6, the reciprocal permittivity vs. temperature is represented. It is observed that, by increasing porosity, the slope of linear fitting decreases, thus indicating a decrease of ferroelectric behaviour [20].

The investigated BZT ceramics closely follow this law above the Curie temperature. The results of Curie–Weiss law fittings are listed in Table 1. When porosity increases, the ferroelectricity decreases due to the progressive reduction of the ferroelectric active phase, as indicated by the decrease of the Curie constant from 1.49 (1% porosity) to 0.92 (21% porosity). However, the Curie temperature remains almost constant irrespective of the porosity, which confirms that the intrinsic nature of the dense component is the same.

### 3.3. High-Field Properties

#### 3.3.1. P(E) Hysteresis Loops

The *P*(*E*) polarization-field loops in the dynamic AC regime have been recorded in order to assess the role of porosity on the switching properties of BZT ceramics. Due to low losses, all ceramic samples display reproducible *P*(*E*) loops. The *P*(*E*) major loops represented in Figure 7a indicate that porosity causes a strong reduction of remanent and saturation polarisation, together with loop tilting that accompanies the reduction of the hysteresis area and the *P_rem_*/*P_sat_* rectangularity loop factor. The polarization at the maximum applied electric field value (8 kV/cm) is in the range of 5–13 µC/cm^2^, with a remanent polarisation in the range of 2–8 µC/cm^2^ as shown in Figure 7b. Both results are in good agreement with other literature data reported for this system [29]. These results can be explained considering that higher pore fraction implies a decrease of the effective electric field on the ferroelectric matrix. This imposes the application of a higher external field in order to induce ferroelectric switching in porous materials. Furthermore, in porous ceramics, the electric field acting on the active material is highly inhomogeneous and smaller than the applied one, as it was demonstrated by finite element calculations in references [9,16]. The observed gradual tilting of the *P*(*E*) hysteresis loops and the reduction of rectangularity with increasing porosity are caused by the local electric field distribution inside the porous material. Large porosity values cause a broadening of the electric field values distribution inside the ferroelectric matrix, thus involving different field values for the local switching to take effect, inducing an inclination in *P*(*E*) loops, as it was demonstrated by calculations in references [9,16].

#### 3.3.2. DC Tunability

After detailed analysis of low-field properties of porous BZT ceramics, we explored the high-field properties of these materials for possible applications as tuneable capacitors. The room temperature *ε*(*E*) dependence (DC tunability) has been determined for increasing/decreasing the DC field. The results are shown in Figure 8a. All ceramics show a strong nonlinearity, with a tendency towards saturation for medium fields of ~10 kV/cm. The samples with 10% and 21% porosity did not endure the application of high fields, most probably due to internal defects. However, after the first increasing/decreasing cycles of the DC field, the nonlinear field dependences *ε*(*E*) tend to stabilize and the collected tunability data are reproducible. Hysteretic behaviour of *ε*(*E*) is reduced with increasing porosity. The tunability of these ceramics, *n* = *ε*(0)/*ε*(*E*), varies from 1.6 (for 1% porosity) to 1.25 (for 19% porosity) at the largest value of the applied electric field (*E* = 10 kV/cm). The tunability values are comparable with the literature-reported ones for similar ceramic compositions [30]. These experimental results confirm the previous FEM calculation results [16], which demonstrated that spherical dielectric pores in a ferroelectric matrix will reduce the permittivity while maintaining a high level of tunability. Even though the increasing porosity reduces the amount of active material (ferroelectric) that causes dielectric nonlinearity, the remaining ferroelectric regions are subjected to much higher field values than in the case of dense material; thus, some tunability is still being maintained.

The 40% reduction of zero-field permittivity, while maintaining a tunability decrease of only 30% at the largest field, together with the strong reduction of hysteresis behaviour, recommend these porous BZT ceramics as suitable materials for tuneable applications.

#### 3.3.3. Piezoelectric Characteristics

Figure 9 shows the variation of the piezoelectric coefficient of BZT ceramics with respect to different porosity levels. It can be observed that the *d*_33_ values decrease from 154 down to 38 pC/N when porosity values increase from 1 to 19%; this corresponds to a 75% decrease when porosity increases to 19%. The decrease of *d*_33_ with increasing porosity is related to: (i) the reduction of piezoelectric fraction per volume unit, which leads to a reduction of remanent polarisation (Figure 7b) and piezoelectric response; and (ii) the reduction of polarisation caused by the concentration of electric field inside the pore region during the poling process, as demonstrated in our previous papers [9,16]. The dense BZT ceramic in this work exhibits a lower *d*_33_ than in other papers due to poling condition: low applied field (only 10 kV/cm, while most papers reported poling at 30–40 kV/cm [29]) and low temperature (the ceramics were poled at room temperature, while the other authors reported 130 °C as poling temperature [31]).

## 4. Conclusions

Porous BaZr_0.15_Ti_0.85_O_3_ ceramics with variable porosity from 5% to 21% have been prepared from solid-state powders using PMMA microspheres as a sacrificial template for inducing 0–3 type microporosity. The role of porosity induced in this way is described in comparison with dense ceramics, i.e., produced without template addition. Large pores in the range of about 15 µm were produced after burning the PMMA microspheres. All the ceramics present pure perovskite structure without any secondary phases. The dielectric data revealed a decrease of permittivity with 40% when porosity increased with 21% from 2600 to 1600. All ceramics show a well-defined ferroelectric–paraelectric phase transition at ~62 °C, followed by Curie–Weiss type behaviour and dielectric losses below 8% in all temperature range. The ferroelectric switching parameters (*P_max_, P_rem_*) and piezoelectric coefficient decrease with increasing porosity as a result of the “dilution” effect. The present study shows that, by PMMA addition, a 0–3 interconnectivity can be induced, and this pore connectivity determines a reduction of zero-field permittivity and hysteretic behaviour of *ε*(*E*), while maintaining a high level of tunability. The study confirms the idea that, by controlling pore interconnectivity and anisotropy, the permittivity and tunability of ceramics can be controlled.

## Figures and Tables

**Figure 1 materials-13-03324-f001:**
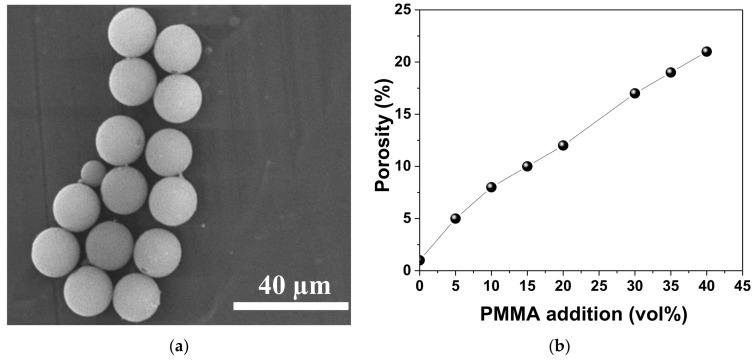
(**a**) SEM image of PMMA microspheres; (**b**) porosity of BZT sintered ceramics as a function of PMMA addition.

**Figure 2 materials-13-03324-f002:**
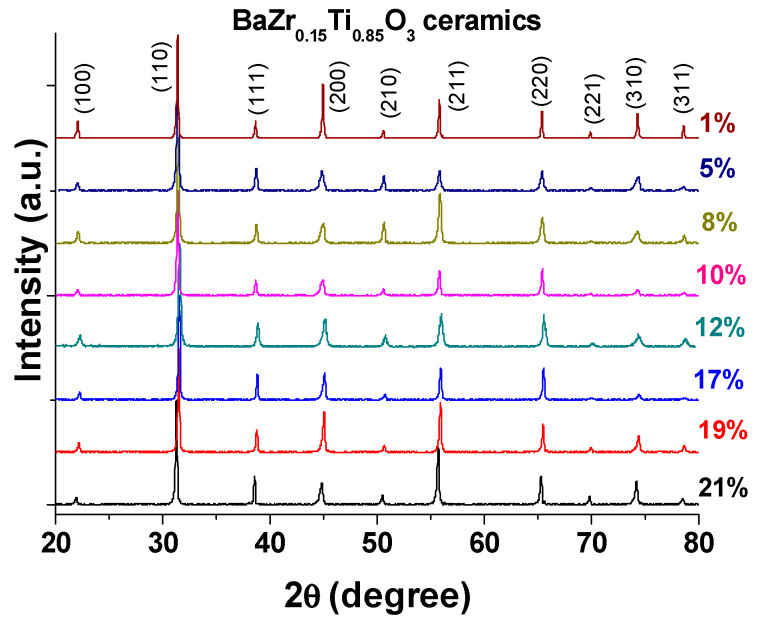
X-ray diffraction patterns of BZT porous ceramics with different porosity levels.

**Figure 3 materials-13-03324-f003:**
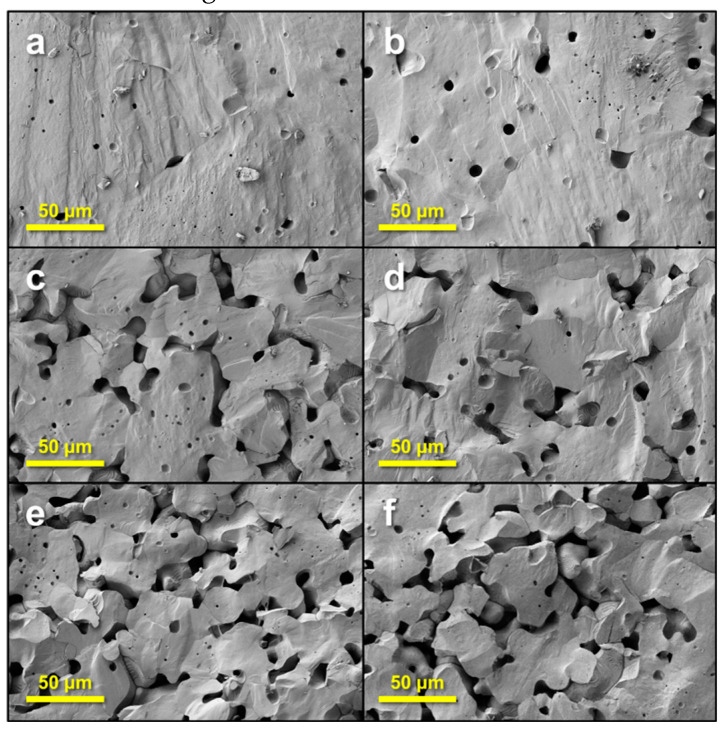
SEM micrographs of porous BZT ceramics with different porosity levels: (**a**) 1%, (**b**) 5%; (**c**) 10%; (**d**) 12%; (**e**) 19%; (**f**) 21%.

**Figure 4 materials-13-03324-f004:**
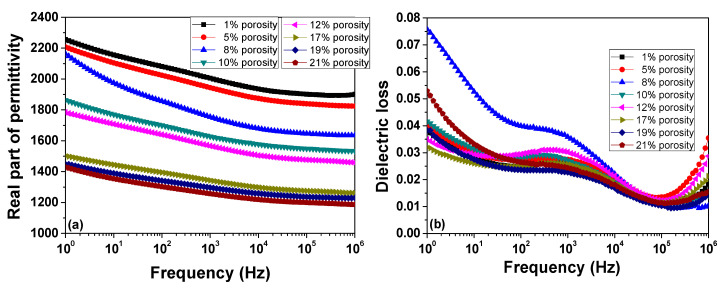
Room temperature frequency dependence of the real part of permittivity (**a**) and of dielectric loss (**b**) for BZT porous ceramics.

**Figure 5 materials-13-03324-f005:**
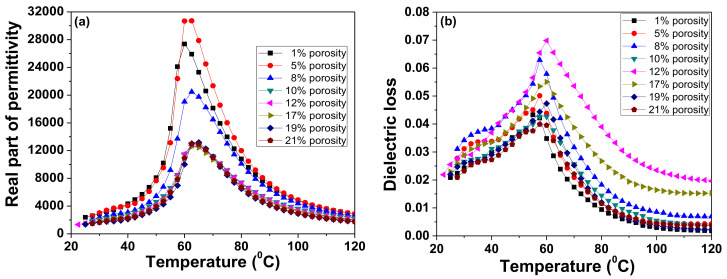
Temperature dependence of the real part of permittivity (**a**) and of dielectric loss (**b**) at 10 kHz.

**Figure 6 materials-13-03324-f006:**
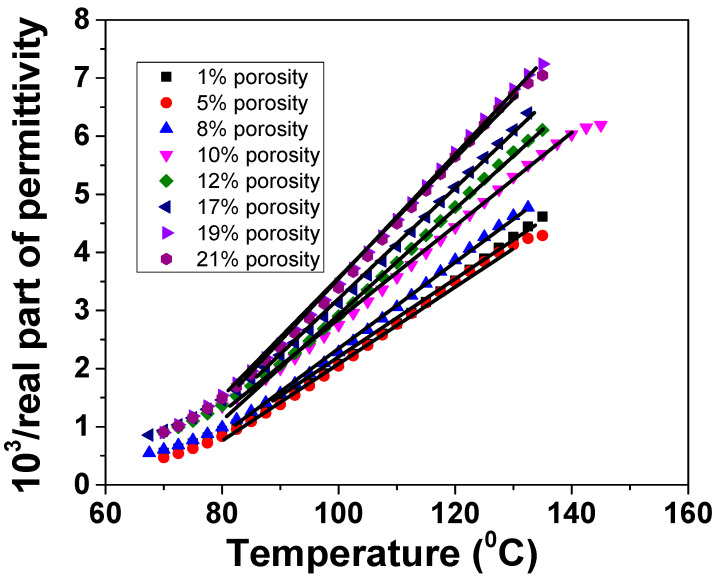
Curie–Weiss analysis for BZT ceramics with different porosities at *f* = 10 kHz.

**Figure 7 materials-13-03324-f007:**
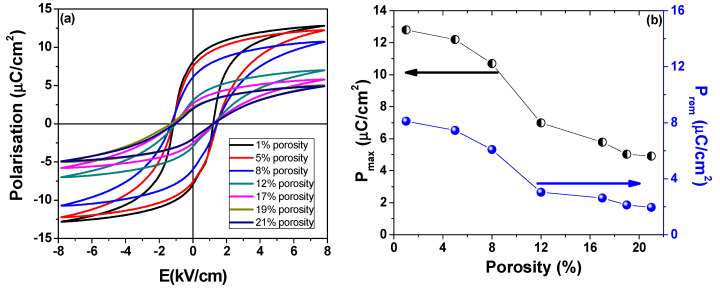
(**a**) Room temperature *P*(*E*) hysteresis loops for BZT porous ceramics; (**b**) ferroelectric switching parameters for BZT porous ceramics extracted from experimental *P*(*E*) hysteresis loops.

**Figure 8 materials-13-03324-f008:**
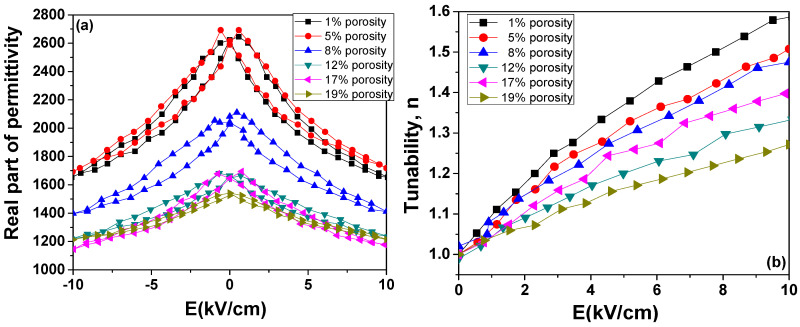
(**a**) Field dependence permittivity at room temperature for BZT porous ceramics; (**b**) DC tunability vs. electric field.

**Figure 9 materials-13-03324-f009:**
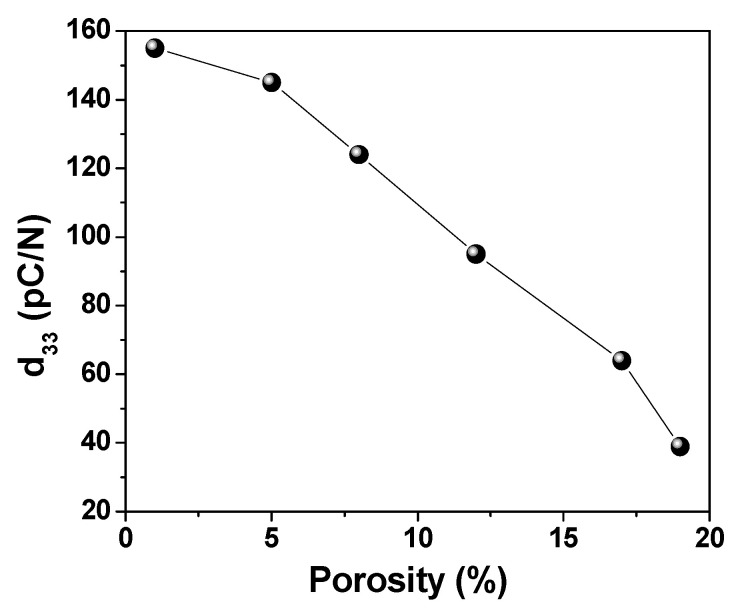
Variation of the piezoelectric coefficient *d*_33_ with porosity level in BZT ceramics.

**Table 1 materials-13-03324-t001:** Curie constants calculated for all BZT porous ceramics.

Sample	*T_m_* (°C)	*T*_0_ (°C)	*ε_m_*	*C* (×10^5^ °C)
1% porosity	60	66	27,363	1.49
5% porosity	62.5	67	30,714	1.45
8% porosity	62.5	67	20,469	1.33
10% porosity	62.5	62	12,972	1.23
12% porosity	62.5	65	12,779	1.12
17% porosity	62.5	66	12,557	1.06
19% porosity	65	67	13,175	0.92
21% porosity	60	69	12,971	0.92

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
