# Peer review of "Effect of Porosity on Functional Properties of Lead-Free Piezoelectric BaZr0.15Ti0.85O3 Porous Ceramics"

_materials, 2020, doi:10.3390/ma13153324_

Round 1

Reviewer 1 Report

The paper reported the preparation and electric properties of porous BaZr0.15Ti0.85O3 ceramics with different porosities by using PMMA as the template. The results are interested in the fields of porous piezoelectric ceramics and related applications. I would like to recommend the publication of the paper after revisions.

The followings are my suggestions to the paper.

  1. Some errors can be found in the paper, for example, “i” is missed in BaZr0.15Ti0.85O3 in the title of the paper. In figure 3, some images used “%” while the others didn’t use.
  2. From the SEM micrographs of figure 3, it is hard to tell the grain size of 200μm and 40μm, while in the XRD data part it says the crystallite size increases from 30.45nm to 49.26nm. On the other hand, the obtained starting BZT powders are around 300nm. A micrograph observation for thermal or chemical etching cross-section is suggested to confirm the grain size.

Author Response

  1. Some errors can be found in the paper, for example, “i” is missed in BaZr0.15Ti0.85O3 in the title of the paper. In figure 3, some images used “%” while the others didn’t use.

We have corrected all writing mistakes.  

  1. From the SEM micrographs of figure 3, it is hard to tell the grain size of 200μm and 40μm, while in the XRD data part it says the crystallite size increases from 30.45nm to 49.26nm. On the other hand, the obtained starting BZT powders are around 300nm. A micrograph observation for thermal or chemical etching cross-section is suggested to confirm the grain size.

We have rephrased this paragraph. Considering the actual conditions (Covid limitations) it was impossible for us to obtained SEM images on thermal etching surface for a better grain size analysis. Also, the discussion of grain size is not essential in our study. The difference between crystallite size and ceramic grains is because all the ceramics are polycrystalline structures.

Reviewer 2 Report

The work by Curecheriu et al. deals with the impact of artificially induced pores in BaZr0.15T0.85O3. Even though the obtained results may, in principle, be quite interesting, the evaluation of the data and especially the presentation remain substandard. An interested but not fully savvy reader does not get a proper introduction or understanding of what the paper is about. This could be due to the numerous grammar and spelling mistakes that make the paper hardly readable. There is one hard to read sentence about the advantages of porous material in ultrasonic applications, and the one given reference is about thin films. Then they suddenly mix it up the tricritical point in BZT. It is not really explained why the composition would need to be at the tricritical point to elucidate the effect of pores on BZT. As the co-existence of the phases does not allow for good structural analysis it is hard to determine any changes which may arise from the synthesis procedure. Even if it was interesting to investigate the material at the TCP, there is no further detail mentioned in the following chapters. The whole discussion and conclusion are not at all based on it. In the results and discussion part, only 6 references are used to compare the work to even though a lot of claims are made, which are not backed-up by the results (see points below). It is unclear what is generally meant by “tunability properties”, the opportunity to tailor the properties of the material by controlling pore structure? If DC-tunabilty is meant, it needs to be explained in much more depth why pores should be beneficial for that. Generally, the DC tunabilty is much higher in dense ceramics (as it is also the case in this work). Only for special pore configuration, the induced field concentration can lead to an increase in DC-tunabilty. This fact is not appropriately discussed or mentioned. The results are also not discussed based on it. In the conclusion, it is just stated that the material can be used for “tunability application” which would basically be false as the DC-tunabilty decreases or it could not be measured as the samples failed. Further critical points are addressed below.

  • There is no reason to assume that Maxwell-Wagner polarization causes the higher loss in the case of the 8% porosity sample as compared to the other samples (from the real part, it becomes clear that it is not the reason). At least it does not become clear from the presented explanations. The discussion is really random and not based on observations. Furthermore, no citations are given for the claims.
  • The authors say that they measured the permittivity at 10 kHz and then state 100 kHz in the caption of Figure 5. What is the reason for this frequency range if it is claimed that this could be the regime for Maxwell-Wagner polarization? Why use only one frequency at all when the capacitance of the contributing processes can be evaluated from the frequency dependent measurements?
  • 8, l. 200 “10% and 21% porosity did not sustain the application of the high fields, most probably due to some internal defects.” It would help to at least know some possible reasons what these defects should be and why these should lead to a breakdown. The whole premise of the paper is pore structure and its impact. For DC-tunabilty it is certainly not beneficial.

For the above-stated reasons, I can unfortunately not support publishing this work in “Materials”.

Author Response

We disagree with the opinion that our work is not worth publishing.  The manuscript has undergone significant improvements and the English language was verified.

  • There is no reason to assume that Maxwell-Wagner polarization causes the higher loss in the case of the 8% porosity sample as compared to the other samples (from the real part, it becomes clear that it is not the reason). At least it does not become clear from the presented explanations. The discussion is really random and not based on observations. Furthermore, no citations are given for the claims.

In composite materials, the increase of permittivity and losses at low frequency is all the time assigned to interfacial polarisation. We added some citations for other porous ceramics with similar behaviour.  

  • The authors say that they measured the permittivity at 10 kHz and then state 100 kHz in the caption of Figure 5. What is the reason for this frequency range if it is claimed that this could be the regime for Maxwell-Wagner polarization? Why use only one frequency at all when the capacitance of the contributing processes can be evaluated from the frequency dependent measurements?

We have corrected the frequency in figure caption.  The Maxwell-Wagner regime is for frequencies lower 100 Hz, and we represented here the temperature dependence at 10 kHz. In this region the permittivity is not affected by extrinsic phenomena.

We have performed measurements for many frequencies, but we presented here only at 10 kHz. If you consider an improvement of our paper, we can add as supplementary materials.  We add here some graphs, and you can observe that for frequency larger than 1 kHz, the frequency dispersion is very low.

  • 200 “10% and 21% porosity did not sustain the application of the high fields, most probably due to some internal defects.” It would help to at least know some possible reasons what these defects should be and why these should lead to a breakdown. The whole premise of the paper is pore structure and its impact. For DC-tunabilty it is certainly not beneficial.

            Sometimes, after many measurements (low field and high field as P(E) loops), ceramics can breakdown. Most of the time this is related with ceramic processing (polish, electrode and cutting). In our case, in tunability experiments the ceramics breakdown took place after the first increasing/decreasing cycles and before stabilizing the response

Reviewer 3 Report

The investigation of some physical properties of perovskite-based systems has been a subject of various studies because of the excellent properties, which makes these materials good candidates for many practical applications. The purpose of the present work was to show the influence of pores on some physical properties of BaZr0.15Ti0.85O3. The manuscript is well written but some minor imperfections that can be improved exist.

  1. Please specify the method by which the porosity of the sample is determined.
  2. The method according to which the grain size was estimated should be described also. Pores occur usually at grain boundaries. Looking at the SEM photos it seems that the size of grains should be smaller than those given in the text. Please comment on it.
  1. Figure 3 - it seems that the SEM photograph with 21% of porosity is missing.
  2. Figure 4. The reciprocal permittivity should be plotted to show how the Weiss-Curie law is fulfilled.
  3. In Table 1 TO - Curie-Weiss temperature should be also added.
  4. Conclusions: the advantages of inducing pores in these ceramics should be highlighted in particular the influence on tunability.
  5. There are several language inaccuracies.

Author Response

  1. Please specify the method by which the porosity of the sample is determined.

The porosity was determined by Arhimedic method.

  1. The method according to which the grain size was estimated should be described also. Pores occur usually at grain boundaries. Looking at the SEM photos it seems that the size of grains should be smaller than those given in the text. Please comment on it.

We have rephrased this paragraph. 

  1. Figure 3 - it seems that the SEM photograph with 21% of porosity is missing.

We added the SEM images for 21% porosity.

  1. Figure 4. The reciprocal permittivity should be plotted to show how the Weiss-Curie law is fulfilled.

We added the figure with Curie-Weiss fitting

  1. In Table 1 TO - Curie-Weiss temperature should be also added.

We added in Table 1 the Curie-Weiss temperature.

Conclusions: the advantages of inducing pores in these ceramics should be highlighted in particular the influence on tunability.

We added a sentence in conclusion paragraph.

  1. There are several language inaccuracies.

The manuscript has undergone significant improvements and the English language was verified.

Round 2

Reviewer 2 Report

The work by Curecheriu et al. deals with the impact of artificially induced pores in BaZr0.15T0.85O3 and still has a lot of interesting results but the scarce changes to the document did not enhance the manuscript. The major points in the review were unfortunately ignored. The Englisch is better but the authors only addressed the minor points of the first review. These comments were not sufficient. Therefore, I cannot change my previous evaluation.

Author Response

Dear reviewer,

We have rephrased the introduction part and we replaced the discussion of tricritical point in BZT. 

The manuscript has undergone significant improvements and the English language was verified. All the modifications and corrections in the new manuscript is indicated in red.

We hope that we have properly addressed all the questions now. 

Best regards,

Lavinia Curecheriu